# Long-Term Simulated Nitrogen Deposition Has Moderate Impacts on Soil Microbial Communities across Three Bioclimatic Domains of the Eastern Canadian Forest

Marie Renaudin [1,2,†], Rim Khlifa [2,3,†], Simon Legault [4], Steven W. Kembel [2], Daniel Kneeshaw [2,3], Jean-David Moore [5] and Daniel Houle [1,2,3,*]

1 Science and Technology Branch, Environment and Climate Change Canada, 105 McGill St., Montréal, QC H2Y 2E7, Canada; enaudin.marie@courrier.uqam.ca
2 Département des Sciences Biologiques, Université du Québec à Montréal, Montréal, QC H2X 1Y4, Canada; khlifa.rim@courrier.uqam.ca (R.K.)
3 Centre d'Étude de la Forêt, Université du Québec à Montréal, Montréal, QC H2X 1Y4, Canada
4 Institut de Recherche en Biologie Végétale, Université de Montréal, Montréal, QC H1X 2B2, Canada
5 Direction de la Recherche Forestière, Forêt Québec, Ministère des Forêts, de la Faune et des Parcs, Québec City, QC G1P 3W8, Canada
* Correspondence: daniel.houle@ec.gc.ca
† These authors contributed equally to this work.

**Abstract:** The soil microbiome plays major roles in the below-ground processes and productivity of forest ecosystems. Atmospheric nitrogen (N) deposition is predicted to increase globally and might create disturbances in soil microbial communities, essentially by modifying soil chemistry. However, the impacts of higher N deposition on the soil microbiome in N-limited northern forests are still unclear. For 16 years, we simulated N deposition by adding ammonium nitrate at rates of 3 and 10 times the ambient N deposition directly into soils located in three bioclimatic domains of the eastern Canadian forest (i.e., sugar maple–yellow birch, balsam fir–white birch, and black spruce–feather moss). We identified changes in the microbial communities by isolating the DNA of the L, F, and H soil horizons, as well as by sequencing amplicons of the bacterial 16S rRNA gene and the fungal ITS region. We found that long-term increased N deposition had no effect on soil microbial diversity, but had moderate impacts on the composition of the bacterial and fungal communities. The most noticeable change was the increase in ectomycorrhizal fungi ASV abundance, potentially due to increased tree root growth on fertilized plots. Our work suggests that, in N-limited northern forests, extra N is rapidly mobilized by vegetation, thus minimizing impacts on the soil microbiome.

**Keywords:** soil microbiome; forest soils; nitrogen atmospheric deposition; temperate forest; boreal forest; ecosystem resilience

## 1. Introduction

The soil microbiome comprises a broad diversity of organisms, including bacteria, fungi, archaea, protists, and viruses, which play crucial roles in biogeochemical cycles and plant productivity, health, and diversity [1,2]. In forest ecosystems, bacteria and fungi are the most abundant microbes in soils [3], and have essential functions such as carbon (C) sequestration, organic matter decomposition, methane oxidation, and nitrogen fixation, which can provide bioavailable nutrients, ultimately promoting plant growth and supporting forest productivity [4–6].

Forest soil microbial communities are strongly influenced by above-ground environmental conditions such as climate and tree species [7,8], and by edaphic factors such as soil moisture, temperature, pH, and nutrient concentrations [9,10]. Nitrogen (N) availability is a particularly important driver of soil bacterial and fungal biomass, diversity, and activity. N can have direct effects on micro-organisms by modifying their substrate stoichiometry

(e.g., soil C:N; [11]) or indirect effects by acidifying soils, subsequently resulting in nutrient leaching [12]. Decreasing soil pH can result in lower phosphorus, magnesium, and calcium availabilities, therefore becoming limiting for microbes and, conversely, leading to aluminum mobilization eventually reaching toxic levels.

While N addition is known to benefit plant and microbial biomass in agricultural systems [13], it can also influence soil microbiomes in unmanaged habitats. Recent meta-analyses have revealed that N deposition is globally detrimental for soil bacterial and fungal biomass and diversity [14–16]. Moreover, N deposition can modify microbial communities' structure by differentially affecting bacterial and fungal taxa. Increasing N deposition was shown to have a negative effect on the relative abundances of arbuscular mycorrhizal fungi and Gram-negative bacteria (e.g., *Acidobacteria* and *Planctomycetes*), but a positive effect on the relative abundances of ectomycorrhizal fungi (ECM), saprophytic fungi, actinomycetes, and Gram-positive bacteria (e.g., *Actinobacteria* and *Firmicutes*) [16–18]. Previous work has also suggested a possible switch of soil bacterial communities according to their life history strategies. Following N fertilization, copiotrophic bacteria would potentially become more abundant than oligotrophic bacteria because they prefer nutrient-rich environments and have a faster growth rate [18,19]. Higher N deposition can also impact below-ground microbiome functions in forests and cause microbial respiration, nitrogen fixation, and methanotrophy to decrease [6,20,21]. Alternatively, it can alleviate microbial N limitation and promote nitrification, denitrification, and methanogenesis [21–23]. Regarding fungal functioning, it has been suggested that N addition could suppress the activity of ligninolytic enzymes in favor of cellulolytic activity [24]. This would translate into a decrease in the abundance of white-rot fungi taxa associated with an increase in brown-rot taxa abundance, thus becoming more efficient at decomposing lignin [25,26].

Since the industrial revolution two centuries ago and the rapid development of human activities, global N deposition has significantly increased [27]. In North America, despite the decrease in the deposition of oxidized forms of N as a result of new governmental environmental policies [28], emissions of reduced inorganic N continued to rise between the 1980s and 2016 in the US [29], potentially impacting North American forests. The influence of N deposition on soil microbes depends on the dose, form, and frequency of the supplies [30,31]. Modification of the N deposition regime can profoundly affect the soil microbiome and, subsequently, forests' vegetation composition and growth [32]. Therefore, understanding the effects of N deposition on the soil microbiome is essential to better estimate and predict the consequences of rising anthropogenic pollution on global nutrient cycles and forest ecosystems' productivity.

The eastern Canadian forest covers more than half of the province of Quebec, representing 2% of the world's forests. It takes part in mitigating climate change as it reduces atmospheric $CO_2$ concentrations by sequestering large amounts of C, mainly in its soils [33]. The eastern Canadian forest is divided into bioclimatic domains, which belong to the northern temperate and boreal vegetation zones and display specific dominating tree and understory plant species. Despite being valuable, most of our knowledge on the effects of N deposition on northern temperate and boreal forests' soil microbiome originates from work conducted in Europe and Asia, and less information is available regarding the eastern Canadian forest [34,35]. Yet, the eastern Canadian forest experiences specific environmental conditions (e.g., climate and forest management practices) that differs from those of other continents [36]. Thus, the impacts of N deposition on soil microbial communities' structure and activity in the eastern Canadian forest may contrast with those in European and Asian forests, and remain to be elucidated. In addition, several studies have focused on determining the influence of N deposition across different forest biomes [14,37], but the impacts of N deposition at the scale of the forest bioclimatic domain are unclear.

In the eastern Canadian forest, N can be a limiting nutrient partially because N deposition is generally low (3–10 kg ha$^{-1}$ yr$^{-1}$) [38], as these ecosystems are often located relatively far from the main anthropogenic N sources (i.e., fossil fuel combustion, agriculture, industrial processes). N deposition may increase in high-latitude forests due to the

expansion of human activities such as logging, mining, and tourism. In these N-limited environments, increasing N deposition could boost microbial activity and forest growth, but in the long-term, these ecosystems could become saturated, and N deposition would have deleterious consequences [39]. However, the exact impacts of long-term rising N deposition on N-limited boreal forests' soil microbial communities remain to be elucidated.

The objective of this study was to investigate the effects of increased N deposition on soil bacterial and fungal communities across three bioclimatic domains in the eastern Canadian forest (i.e., sugar maple–yellow birch, balsam fir–white birch, and black spruce–feather moss) receiving contrasting natural N deposition. Based on previous results, we hypothesized that increased N deposition would have important impacts on soil microbial communities' structure and diversity, and that these impacts would vary according to spatial location (i.e., study sites and soil horizons). We predicted that, under elevated N deposition conditions, (i) soil microbial alpha-diversity would decrease [15], (ii) copiotrophic bacteria would become more abundant than oligotrophic bacteria [18,19], and (iii) fungal communities would switch from white-rot taxa-dominated to brown-rot taxa-dominated [25,26]. We simulated long-term supplementary N deposition by adding exogenous ammonium nitrate at the surface of the experimental sites' soil for 16 years. Then, we assessed the impacts of N deposition on the bacterial and fungal relative abundance and diversity by extracting the DNA of 78 samples of L, F, and H soil organic horizons and by sequencing amplicons of the bacterial 16S rRNA gene and the fungal ITS region.

## 2. Materials and Methods

### 2.1. Site Description

Soil samples were collected at three forest sites within distinct bioclimatic domains between 46° N and 49° N in the province of Québec, Canada (Table S1). Lac Clair, the southernmost experimental site, is located in the northern temperate zone and belongs to the sugar maple–yellow birch domain, where the vegetation is dominated by sugar maple (*Acer saccharum* Marsh.), beech (*Fagus grandifolia* Ehrh.), and yellow birch (*Betula alleghaniensis* Britt.) trees. The Lac Laflamme and Lac Tirasse sites are both located in the boreal zone and belong to the balsam fir–white birch and black spruce–feather moss domains, respectively. *Abies balsamea* (L.) Mill. is the dominating tree species at Lac Laflamme, whereas *Picea mariana* (Mill.) B.S.P. is the dominating tree species at Lac Tirasse. At these two sites, the understory vegetation mostly includes mosses such as *Pleurozium schreberi* (Brid.) Mitt. and *Ptilium crista-castrensis* (Hedw.) De. Not, as well as ericaceous shrubs (e.g., *Rhododendron* spp., *Vaccinium* spp.). Hereafter, sites will be referred to by the dominating tree species in the forest bioclimatic domain to which they belong.

The soils of the experimental sites are classified as Orthic Humo-Ferric Podzols, and Typic Haplorthod at the balsam fir site [40]. The organic horizons are acidic, with pH levels of 2.9–4.1. The mean annual air temperature is 3.4 °C at the sugar maple site, −0.6 °C at the balsam fir site, and 0 °C at the black spruce site, whereas the mean annual soil temperature is relatively similar among the sites, ranging between 3.6 °C and 5 °C. As natural atmospheric N deposition decreases with latitude in Quebec [38], it is relatively low to moderate in the boreal zone sites (3 kg ha$^{-1}$ yr$^{-1}$ at the black spruce site and 5.7 kg ha$^{-1}$ yr$^{-1}$ at the balsam fir site), but reaches 8.5 kg ha$^{-1}$ yr$^{-1}$ at the sugar maple site, the southernmost site [41]. However, these rates of natural N deposition remain low compared to other experimental forest sites located in the eastern U.S. and Europe [42].

During the 16-year-long experiment, no major forest disturbances such as insect outbreaks, fires, or windfalls, were observed at the study sites. Additional information regarding the environmental characteristics of the three experimental sites can be found in Table S1.

### 2.2. Experimental Design, Sampling Procedure, and Chemical Analyses

At each site, 9 experimental units of 15 m × 15 m (at the sugar maple site) or 10 m × 10 m (at the balsam fir and the black spruce sites), separated by at least 10–15 m,

were delimited. Treatments (control, "low N", and "high N" additions) were randomly assigned to blocks of three experimental unit replicates. The low N addition treatment represented a 3-fold increase in the natural atmospheric N deposition rate received by each forest site at the beginning of the experiment (i.e., applications of 9 kg ha$^{-1}$ yr$^{-1}$ at the black spruce site, 17 kg ha$^{-1}$ yr$^{-1}$ at the balsam fir site, and 26 kg ha$^{-1}$ yr$^{-1}$ at the sugar maple site), while the high N addition treatment represented a 10-fold increase (i.e., applications of 30 kg ha$^{-1}$ yr$^{-1}$ at the black spruce site, 57 kg ha$^{-1}$ yr$^{-1}$ at the balsam fir site, and 85 kg ha$^{-1}$ yr$^{-1}$ at the sugar maple site). The N treatments consisted of additions of a commercial $NH_4NO_3$ solution supplied in four passes using a backpack sprayer (Solo, Newport News, VA, USA) to ensure even applications. Starting in June 2001, treatments were performed monthly, 5 times per year, during the summer (from June to October).

Samples of L (litter), F (fermented), and H (humic) soil organic horizons [40] were collected in early September 2017 at the black spruce site and in early October 2017 at the balsam fir site and the sugar maple site. At the black spruce site, litter was absent from the high N treatment units and was, therefore, not collected. For each experimental unit, three samples per organic horizon were collected and pooled into a single composite sample, resulting in 3 replicated samples per experimental block. All samples were collected with a soil corer, placed in sterile plastic bags, and stored at −20 °C. In total, 78 samples were collected and used for DNA analysis.

Additionally, samples of the F and H horizons were collected in October 2013 and 2021 at each site, and soil pH, N concentration, and C:N were measured after 12 and 20 years, respectively, of N fertilization. These sampling dates correspond to 4 years before and 4 years after the soil sampling date for DNA analysis, and allowed us to estimate whether soil chemistry was modified by N addition treatments. Following the same procedure as described previously, 2 replicates per experimental unit were sampled and stored at room temperature. Soil samples were then air-dried, ground, and sieved to 2 mm. The total C and N concentrations were measured by dry combustion (LECO model TruMac). The soil pH was measured with a probe following the procedure described in Moore and Houle (2023) [43].

### 2.3. DNA Isolation, Amplicon Sequencing, and Taxonomy

Samples were partially unfrozen at room temperature for a short period of time to enable their homogenization before DNA extraction. Larger plant debris in the litter horizon (e.g., tree leaves, needles, and twigs) was cut into smaller pieces with sterile scissors, and H and F horizon samples were processed through sterile 5 mm and 2 mm sieves. Then, 2 g of each homogenized sample was placed into a bead beater for 1 min and stored at −80 °C until DNA extraction.

The total DNA was extracted from 0.25 g of each sample using the DNeasy PowerSoil kit (Qiagen, Hilden, Germany), following the manufacturer's protocol. DNA concentration and quality were verified using a NanoDrop spectrophotometer (Thermo Fisher Scientific, Waltham, MA, USA). Then, the V5–V6 region of the bacterial 16S rRNA gene and the fungal ITS region were amplified by PCR using the primer pairs 799F/1115R [44,45] and ITS1F/ITS2R [46,47], respectively. All PCR reactions consisted of 1 μL sample DNA, 5 μL 5 × HF buffer (Thermo Fisher Scientific), 0.5 μL dNTP (10 μM), 0.5 μL forward and reverse primers (10 μM), 0.75 μL DMSO, 0.25 μL Phusion HotStart II polymerase (Thermo Fisher Scientific), and 16.5 μL water (molecular grade). Both gene regions were amplified by 30 s of denaturation at 98 °C followed by 35 cycles of 15 s at 98 °C, 30 s at 64 °C, 30 s at 72 °C, and a final elongation step of 10 min at 72 °C. Positive and negative controls were used for each PCR run. Then, PCR products were cleaned and normalized with the SequalPrep Kit (Thermo Fisher Scientific), and were subsequently purified using Agencourt AMPure XP beads (New England Biolabs, Ipswich, MA, USA). Multiplexed libraries were prepared using equimolar concentrations of DNA, and amplicons were sequenced on the Illumina MiSeq 250 bps platform (Illumina, San Diego, CA, USA) at Genome Quebec (Montreal, QC, Canada).

Raw sequence files were demultiplexed using QIIME software version 1.9.1. [48], and were subsequently processed following the DADA2 pipeline (package dada2 version 1.8.0.; [49]) using the R software version 4.2.1. [50]. To eliminate low-quality reads, sequences were filtered and truncated at position 210 for the forward reads and at position 157 for the reverse reads. Then, paired-ended reads were merged, chimeric sequences were removed, and ASVs (amplicon sequence variants) were determined. Singletons, as well as non-bacterial and non-fungal ASVs, were discarded. Using the phyloseq package, version 1.40.0. [51], samples were rarefied to the minimum number of sequences per sample (i.e., 13,502 sequences for bacteria and 2716 sequences for fungi) [52]. Finally, taxonomy was assigned using the SILVA database version 123 for 16S rRNA [53] and the UNITE database version 8.2 for ITS [54].

In the discussion, the suggested bacterial functional classification was based on the work of Fierer et al. (2007) [55], and the suggested fungal functional classification (i.e., primary and secondary lifestyle) was based on the FungalTraits version 1.2. database [56].

*2.4. Statistical Analyses*

All statistical analyses were performed using R software, and the figures were built using the ggplot2 package, version 3.3.6. [57].

First, we examined the global composition of the microbial communities by calculating the relative abundance of the top 20 bacterial and fungal genera for each set of replicates with the tidyr package, version 1.2.0. [58], and we estimated alpha-diversity by calculating the Shannon index with the phyloseq package. Significant differences in microbial relative abundance, microbial diversity, soil N concentration, C:N, and pH between sites, treatments, and soil horizons were assessed using ANOVA and post-hoc Tukey's tests when data were normally distributed, or Kruskal–Wallis and Dunn's post hoc tests (adjusted with the Benjamini–Hochberg method) when data were not normally distributed. Normality was assessed using the Shapiro–Wilk test.

To further assess the effects of different sites, soil horizons, and treatments on soil bacteria and fungi, we performed non-metric multidimensional scaling (NMDS) ordination and PERMANOVA on Bray–Curtis dissimilarities with the vegan package, version 2.6-2 [59]. For PERMANOVAs, Bray–Curtis dissimilarities were calculated on normalized data counts using variance stabilizing transformation [60] with the phyloseq and DESeq2 packages, version 1.36.0. [61].

Finally, to detect taxa whose abundance differed significantly among groups of samples, we performed ASV differential abundance analyses on unrarefied data between sites, soil horizons, and treatments using the DESeq2 package as well.

## 3. Results

*3.1. Characterization of Soil Bacterial and Fungal Communities in L, F, and H Horizons from Three Contrasted Forest Bioclimatic Domains*

From the 78 forest soil samples which were collected, we obtained a total of 3,992,992 16S rRNA raw sequences and 2,061,905 ITS raw sequences. After quality filtering and cleaning (i.e., removal of singletons as well as chimeric, non-bacterial, and non-fungal sequences) and the exclusion of one sample due to contamination, a total of 2,242,773 16S rRNA sequences and 1,374,990 ITS sequences remained. The number of filtered sequences per sample ranged between 13,502 and 55,336 for 16S rRNA and between 2716 and 41,223 for ITS. After rarefaction, a total of 12,492 unique ASVs from 1,039,654 16S rRNA sequences and 3960 unique ASVs from 209,132 ITS sequences were detected and used for subsequent analyses.

Bacterial communities were composed of 26 phyla, 49 classes, and 358 genera. The five most abundant bacterial phyla were, in this order, *Proteobacteria* (42.2%), *Actinobacteria* (28.7%), *Acidobacteria* (24.2%), *Bacteroidetes* (2.2%), and *Armatimonadetes* (0.6%), whereas the five most abundant genera were *Acidothermus* (16.7%), *Mycobacterium* (6.7%), *Granulicella* (6.5%), *Roseiarcus* (6.4%), and *Bradyrhizobium* (5.7%). The fungal communities

were composed of 11 phyla, 37 classes, and 397 genera. The five most abundant fungal phyla were Basidiomycota (49.3%), Ascomycota (45.6%), Mortierellomycota (4.8%), Mucoromycota (0.1%), and Rozellomycota (0.07%), whereas the five most abundant genera were *Russula* (11.7%), *Piloderma* (8.8%), *Meliniomyces* (8.7%), *Archaeorhizomyces* (6.2%), and *Mortierella* (6%).

The sampling site had a strong effect on the soil microbial communities, and explained 17.7% of the bacterial communities' structures and 25.1% of the fungal communities' structures, thus being ranked as the second-best driver and the best driver, respectively (Table 1).

**Table 1.** PERMANOVA analysis of Bray–Curtis dissimilarities, showing the variation in forest soil bacterial (a) and fungal (b) communities' structure. These are explained by the sampling site, treatment (control, low N addition, or high N addition), soil horizon, and their interactions.

| | (a) Bacterial Communities [b] | | | | (b) Fungal Communities [c] | | | |
|---|---|---|---|---|---|---|---|---|
| | **Df** | **F** | **$R^2$** | **$p$ [a]** | **Df** | **F** | **$R^2$** | **$p$ [a]** |
| Site | 2 | 18.36 | 0.177 | 0.001 * | 2 | 18.14 | 0.251 | 0.001 * |
| Horizon | 2 | 32.81 | 0.316 | 0.001 * | 2 | 8.40 | 0.116 | 0.001 * |
| Treatment | 2 | 2.21 | 0.021 | 0.006 * | 2 | 1.70 | 0.023 | 0.012 * |
| Site × Horizon | 4 | 7.42 | 0.143 | 0.001 * | 4 | 5.40 | 0.149 | 0.001 * |
| Site × Treatment | 4 | 1.55 | 0.030 | 0.032 * | 4 | 1.62 | 0.045 | 0.002 * |
| Horizon × Treatment | 4 | 1.45 | 0.028 | 0.049 * | 4 | 0.84 | 0.023 | 0.861 n.s. |
| Site × Horizon × Treatment | 7 | 1.11 | 0.038 | 0.215 n.s. | 7 | 0.83 | 0.040 | 0.954 n.s. |

[a] *, $p < 0.05$; n.s., not significant. [b] The model explains 75.3% of the variation in the bacterial community structure. [c] The model explains 64.7% of the variation in the fungal community structure.

Across the three sites, there were noticeable changes in the major microbial taxa ASV and their relative abundances. The ASV abundances of the bacterial phyla *Acidobacteria*, *Proteobacteria*, and *Chloroflexi* were higher at the sugar maple site than at the other two sites (Figure S1a). The black spruce site had the highest *Actinobacteria* ASV abundance, but also the lowest *Acidobacteria* and *Proteobacteria* ASV abundances. At the genus level, the relative abundances of *Galbitalea* (in the L horizon; $\chi^2 = 6.49$, $p = 0.03$), *Gemmatirosa* (in the F horizon; $\chi^2 = 6.49$, $p = 0.03$ and H horizon; $\chi^2 = 7.20$, $p = 0.02$), and *Leptothrix* (in the L horizon; $\chi^2 = 7.20$, $p = 0.02$) were higher at the sugar maple site than at the black spruce site (Figure 1a).

On the contrary, the relative abundance of *Acidipila* was lower at the sugar maple site than at the black spruce site (F horizon; $\chi^2 = 7.20$, $p = 0.02$). The balsam fir site also displayed a higher relative abundance of *Mycobacterium* than the sugar maple site (F horizon; $\chi^2 = 6.49$, $p = 0.03$). Regarding fungal communities, the sugar maple site displayed the highest ASV abundance of phyla Ascomycota and Mortierellomycota, and the black spruce site displayed the highest ASV abundance of Basidiomycota (Figure S1b). However, the black spruce site also had the lowest ASV abundance of Mortierellomycota, the balsam fir site had the lowest abundance of Ascomycota, and the sugar maple site had the lowest abundance of Basidiomycota. Compared to bacteria, there were more differences in the relative abundances of fungal genera between sites, which was supported by the fact that the site was a better driver of fungal communities' variation than bacterial communities' variation. For the sugar maple site, the major differences included higher relative abundances of *Meliniomyces* (L horizon; $\chi^2 = 6.16$, $p = 0.04$ and F and H horizons; $\chi^2 = 7.20$, $p = 0.02$), *Pezoloma* (L horizon; $\chi^2 = 7.2$, $p = 0.02$ and F horizon; $\chi^2 = 6.71$, $p = 0.03$), and *Russula* (L horizon; $\chi^2 = 6.71$, $p = 0.03$) compared to the black spruce site, and lower relative abundances of *Tylospora* (L and F horizons; $\chi^2 = 7.62$, $p = 0.02$) compared to the balsam fir site (Figure 1b). The relative abundance of *Tomentella* was higher at the balsam fir site than at the black spruce site (L horizon; $\chi^2 = 6.76$, $p = 0.03$) but the opposite was true for the relative abundance of *Archaeorhizomyces* (F horizon; $\chi^2 = 6.16$, $p = 0.04$). These abundance results are in accordance with the ordination plots, which show that the

microbial communities are more distinct at the sugar maple site than at the other sites (especially the black spruce site; Figure 2).

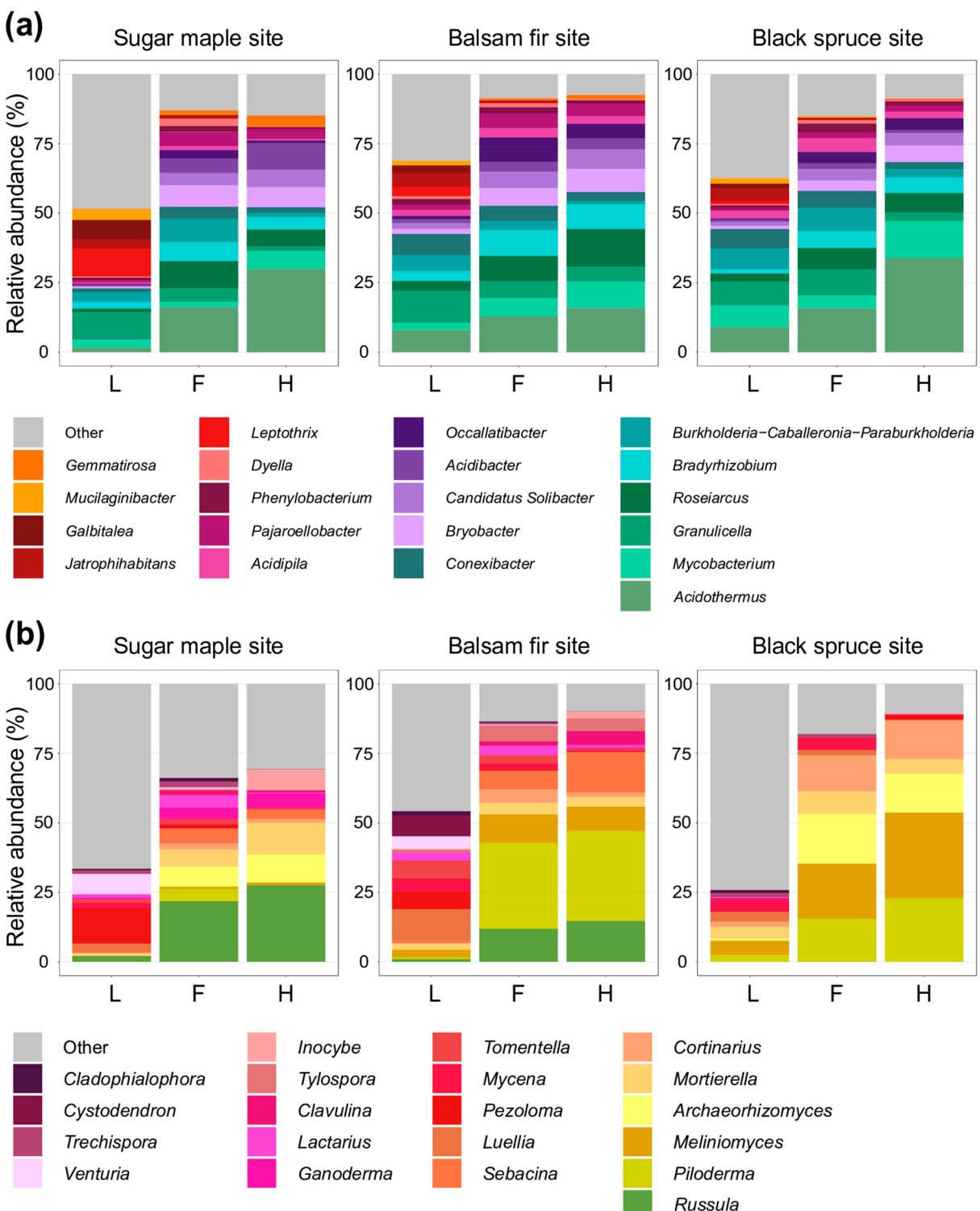

**Figure 1.** Relative abundances of bacterial (**a**) and fungal (**b**) dominant genera in the L, F, and H soil horizons collected on control plots at the sugar maple, balsam fir, and black spruce sites (*n* = 9).

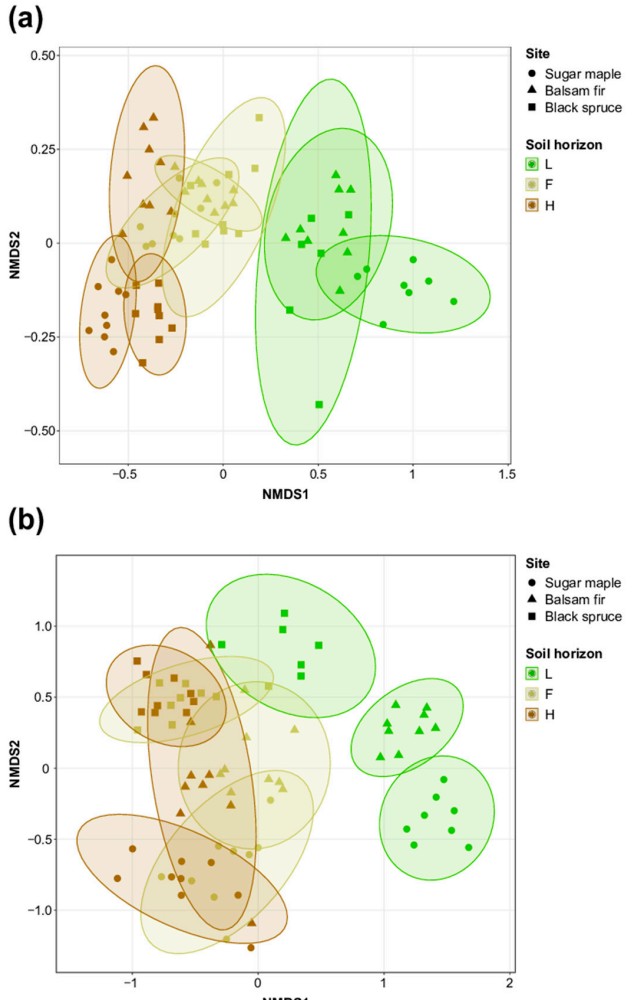

**Figure 2.** Non-metric multidimensional scaling (NMDS) ordination on Bray–Curtis dissimilarities showing structure variation in bacterial (**a**) and fungal (**b**) communities from the L, F, and H soil horizons collected at the sugar maple, balsam fir, and black spruce sites. Each NMDS regroups 77 observations. Points represent samples, and ellipses represent 95% confidence intervals. Point shapes were assigned to study sites, and colors were assigned to soil horizons.

The bacterial alpha-diversity (estimated with Shannon index) was higher than the fungal alpha-diversity for all sites and soil horizons (Figure 3). Alpha-diversity in the L horizon was significantly lower at the black spruce site than at the balsam fir site (F = 5.69, $p = 0.04$ for bacteria; $\chi^2 = 6.49$, $p = 0.03$ for fungi). However, no significant differences in diversity were detected for the F and H horizons between sites.

The soil horizon also had an important effect on the bacterial and fungal communities. The horizon explained 31.6% of the bacteriome variation, therefore being ranked as the best driver of bacterial communities' structures, and 11.6% of the mycobiome variation, therefore being ranked as the third-best driver of fungal communities' structures (Table 1). There was a significant contribution made by the interaction between the soil horizon and sampling site to the variation in the microbial communities (~14% for both bacteria and fungi), which suggests that the effects of the soil horizon on the microbiome's structure also depend on the site under consideration. At the ASV level, the L horizon had higher abundance of *Actinobacteria*, *Proteobacteria*, *Bacteroidetes*, and *Chloroflexi*, but a lower abundance of *Acidobacteria* and *Verrucomicrobia*, compared to the F and H horizons (Figure S2a). The main changes in the relative abundances of genera included a higher relative abundance of *Galbitalea* (at all sites: $\chi^2 = 6.49$–7.45, $p = 0.02$–0.03), *Leptothrix* (at all sites: $\chi^2 = 7.26$–7.45, $p = 0.02$), and *Jatrophihabitans* (at all sites: $\chi^2 = 6.16$–7.45, $p = 0.02$–0.04) in litter compared to

the H horizon, and higher relative abundances of *Acidothermus* (at all sites: $\chi^2$ = 6.49–7.2, $p$ = 0.02–0.03) and *Acidibacter* (at the sugar maple site only: $\chi^2$ = 7.2, $p$ = 0.02) in the H horizon compared to in litter (Figure 1a). For the fungal communities, the ASV abundance of the Ascomycota and Basidiomycota phyla were higher in the litter than in the F and H horizons. On the contrary, the ASV abundance of Mortierellomycota was lower in the litter than in deeper horizons (Figure S2b). The relative abundance of the *Cystodendron* genus (balsam fir site; $\chi^2$ = 7.62, $p$ = 0.02) was higher in the litter than in the F horizon, whereas the relative abundances of *Cladophialophora* (balsam fir site: $\chi^2$ = 6.54, $p$ = 0.03 and black spruce site: $\chi^2$ = 6.49, $p$ = 0.03), *Pezoloma* (sugar maple site: $\chi^2$ = 7.45, $p$ = 0.02 and balsam fir site: $\chi^2$ = 7.2, $p$ = 0.02), *Luellia* (sugar maple site: $\chi^2$ = 7.62, $p$ = 0.02 and balsam fir site: $\chi^2$ = 6.72, $p$ = 0.04), and *Venturia* (sugar maple and balsam fir sites: $\chi^2$ = 7.45, $p$ = 0.02) were higher in the litter than in the H horizon (Figure 1b). No genus was significantly more abundant in the F-H horizons compared to L, but the relative abundances of *Cortinarius*, *Piloderma*, and *Russula* tended to increase with the horizon depth at the three sites. These abundance results are supported by ordination plots showing that the microbial compositions of the F and H horizons were more similar than the microbial composition of the L horizon; this observation was even more pronounced for fungi (Figure 2). The alpha-diversity of both the bacterial and fungal communities tended to decrease with the horizon depth (Figure 3). The bacterial diversity significantly decreased between the litter and H horizon at the balsam fir site ($\chi^2$ = 8.85, $p$ = 0.01), whereas fungal diversity significantly decreased between the litter and F and H horizons at the sugar maple site (F = 7.39, $p$ = 0.03 and 0.04) and at the balsam fir site (F = 49.42, $p$ = 0.001 and $1.7 \times 10^{-4}$).

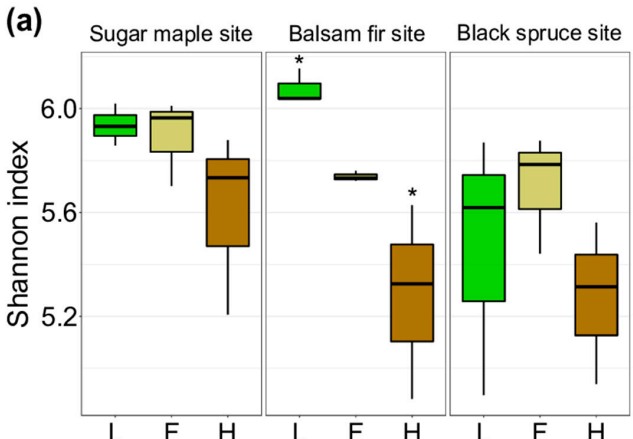

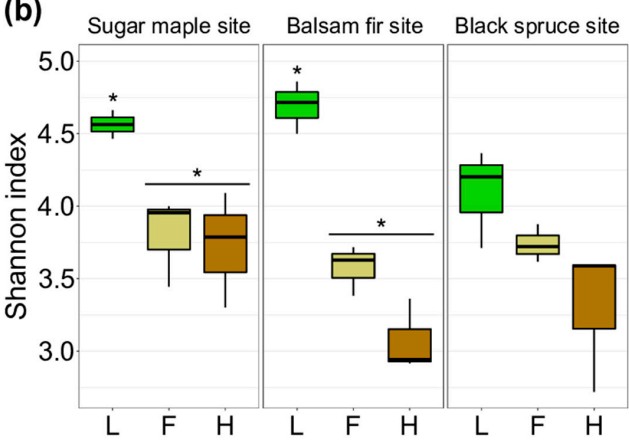

**Figure 3.** Bacterial (**a**) and fungal (**b**) alpha-diversity (Shannon index) in the L, F, and H soil horizons collected on control plots at the sugar maple, balsam fir, and black spruce sites (*n* = 9). Stars indicate statistically significant differences between soil horizons' diversity levels at each site.

### 3.2. Impacts of Nitrogen Addition on Soil Microbiome and Chemistry at the Site and at the Horizon Level

N treatment had a significant effect on the soil microbial communities, but it was clearly less important than the influence of the sampling site and soil horizon (Table 1). N treatment explained 2.1% of the bacteriome variation and 2.3% of the mycobiome variation. The ASV differential abundance analysis revealed that N addition had significantly fewer effects on bacterial than on fungal abundance, and that, overall, N addition had negative impacts on bacterial abundance, but positive impacts on fungal abundance (Figure 4).

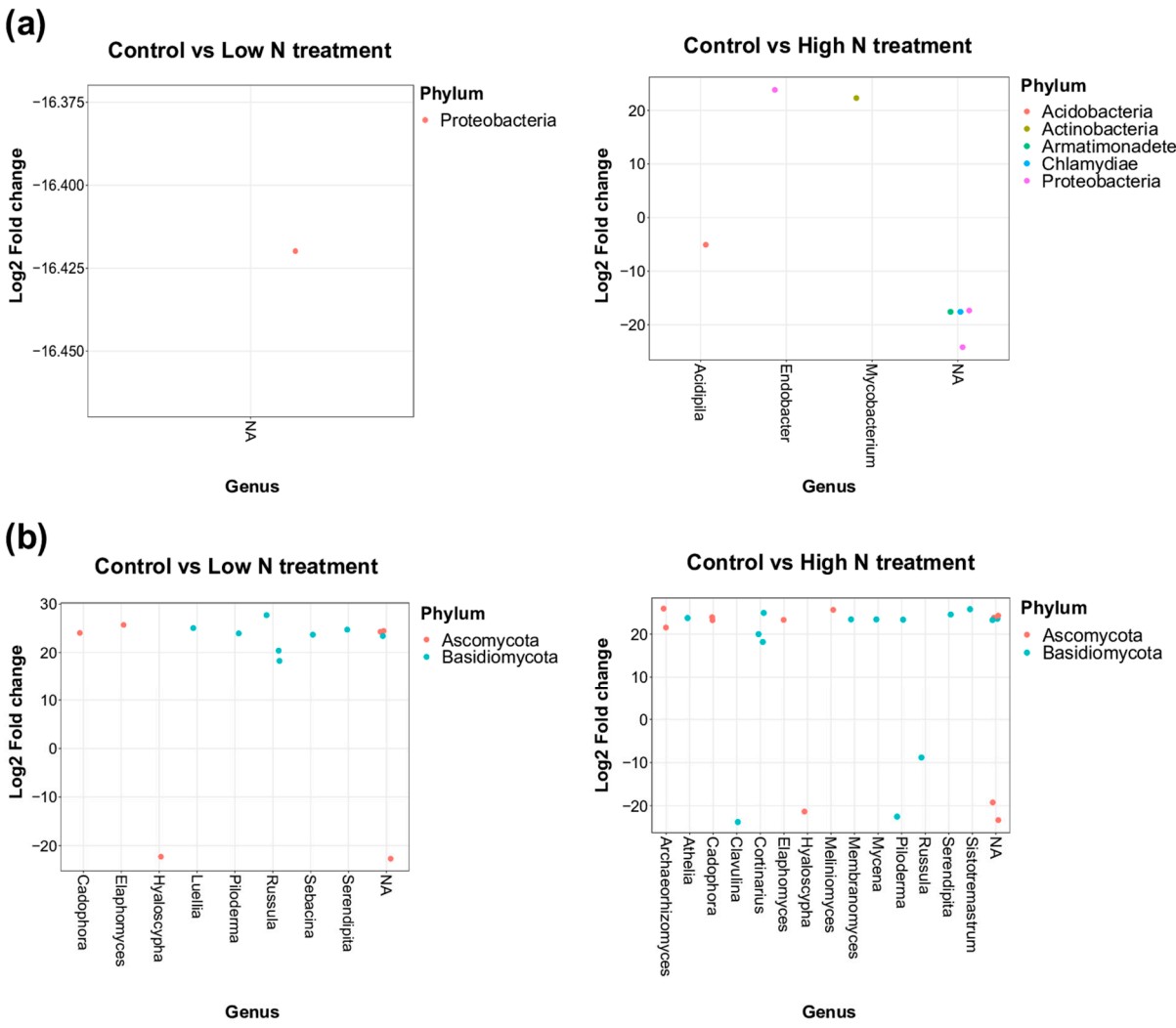

**Figure 4.** Soil bacterial (**a**) and fungal (**b**) ASV differential abundance analysis between control, low nitrogen (N), and high N treatment plots located at the sugar maple, balsam fir, and black spruce sites. Points represent ASVs with abundances differing significantly ($p < 0.05$) between treatments. Colors were assigned to phyla.

For bacteria, low N addition only led to a decrease in *Proteobacterial* ASV, while high N addition only decreased *Acidipila* ASV abundance and increased the *Endobacter* and *Mycobacterium* ASV abundances. For fungi, N addition only affected the Ascomycota and Basidiomycota phyla. Among the most abundant genera, low N addition increased the *Luellia*, *Piloderma*, and *Sebacina* ASV abundance and high N addition increased the *Archaeorhizomyces*, *Cortinarius*, *Membranomyces*, *Meliniomyces*, *Serendipita*, *Sistotremastrum*, and *Mycena* ASV abundance. Both treatments negatively affected the ASV abundance of *Hyaloscypha*, and high N addition decreased the abundance of *Clavulina* and *Russula.* The

microbial relative abundances in the control and N addition treatments were very similar (Figure 5).

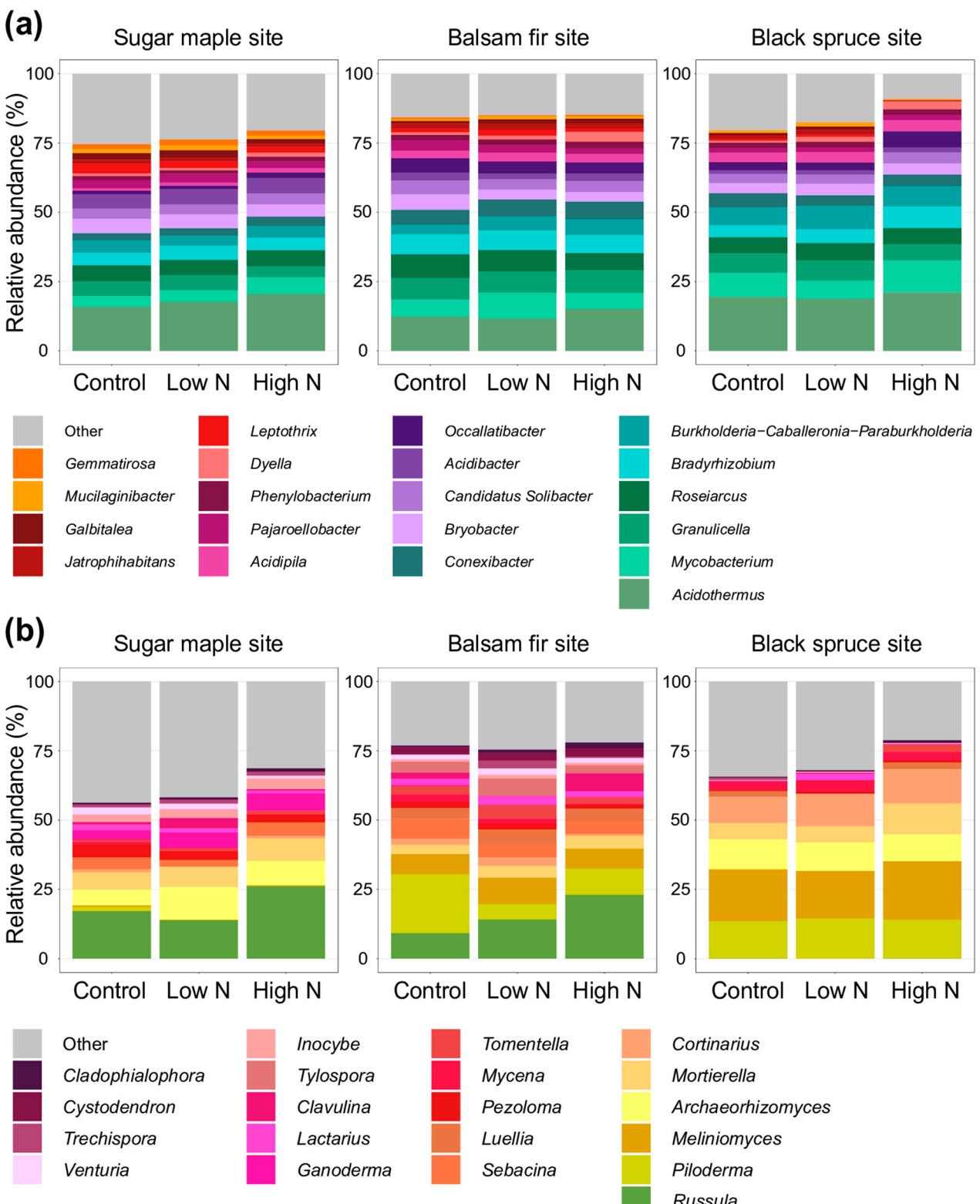

**Figure 5.** Relative abundance of bacterial (**a**) and fungal (**b**) dominant genera in soils collected in the control, low nitrogen (N), and high N treatment plots at the sugar maple, balsam fir, and black spruce sites (*n* = 9).

The most noticeable significant change was the higher relative abundance of the fungal genus *Tomentella* under high N addition at the black spruce site ($\chi^2$ = 15.38, $p$ = 0.001). Finally, low and high N additions did not have a significant impact on either bacterial or fungal diversity at any of the three sites (Figure 6).

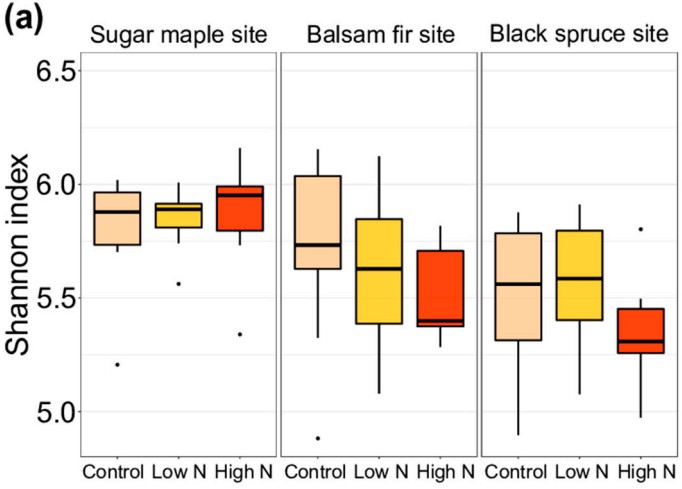

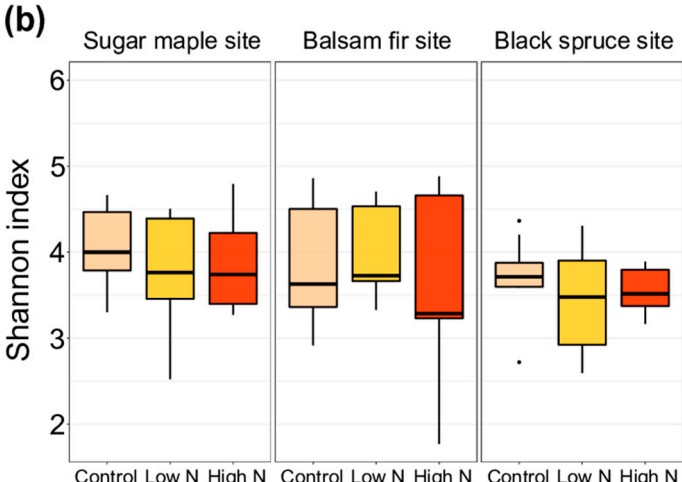

**Figure 6.** Bacterial (**a**) and fungal (**b**) alpha-diversity (Shannon index) in soils collected in the control, low nitrogen (N), and high N treatment plots at the sugar maple, balsam fir, and black spruce sites (*n* = 9).

The concentrations of N, C:N, and pH in the soil horizons of the study sites were either not significantly (H horizon) or only weakly (F horizon) impacted by chronic N addition (Table S2). In 2013, the pH of horizon F was significantly lower under high N treatment (mean pH = 2.86) compared to the control (mean pH = 3.04) at the sugar maple site (F = 6.46, $p$ = 0.0367; Figure S3). At the balsam fir site, also in 2013, the mean N concentration of horizon F was 21.60 mg·g$^{-1}$ under high N treatment, which was significantly higher than the control (mean of 18.50 mg·g$^{-1}$; F = 9.08, $p$ = 0.0217) and low N treatment (mean of 18.63 mg·g$^{-1}$; F = 9.08, $p$ = 0.0262). On the contrary, the C:N was significantly lower under high N conditions (mean of 26.33 mg·g$^{-1}$) compared to the two other treatments (mean of 31.46 mg·g$^{-1}$, F = 8.59, $p$ = 0.0242 for the control; mean of 31.21 mg·g$^{-1}$, F = 8.59, $p$ = 0.0297 for low N treatment).

## 4. Discussion

### 4.1. Spatial Location as the Main Driver of Forest Soil Microbial Community Composition

We demonstrated that the sampling site and the soil horizon were the two variables with the strongest influence on forest soil bacterial and fungal communities, as, together, they explained more than 36% of the microbial communities' structural variation. The soil horizon was the most important driver of bacterial communities, while the site was the most important driver of fungal communities (Table 1). On the other hand, N addition had a significant, but very low, effect on the soil microbiome, explaining less than 2.5% of microbial communities' structure variation. Moreover, the ASV differential abundance analyses, taxa relative abundance, and alpha-diversity results indicated that N addition had minor effects on bacteria and fungi, and that these effects were positive overall at the ASV level (Figure 4). This is contradictory to our hypothesis, which stated that higher N deposition would have substantial impacts on soil microbes. Instead, we found that the compositions of the soil bacterial and fungal communities shifted vertically with space on a large scale (sampling sites), but also on a fine scale, within the soil (soil horizons).

Soil bacterial and fungal community composition and diversity are known to vary greatly with geographic location, as they are strongly influenced by their environment. The three study sites selected for this work were located in distinct bioclimatic domains and were, therefore, exposed to different habitat conditions (e.g., air and soil temperature, soil pH, vegetation composition, C:N; Table S1) which constitute drivers of the structure, biomass, and activity of soil microbial communities [9,62]. In particular, tree stand identity has an important effect on the forest soil microbiome, mostly because of its tight control over soil pH and nutrient content through litter production [63,64]. Soil pH and nutrients (e.g., C, N, P) are considered to be the main factors regulating bacterial growth and activity in soil [3], but the nature and intensity of their effects are taxa-dependent. For example, *Actinobacteria* were shown to be more abundant in forest soils with higher C:N, whereas *Acidobacteria* were found to be more abundant in acidic forest soils [10,63,65]. Similarly, we found that *Actinobacteria* were more abundant at the site with the highest soil C:N (i.e., the black spruce site). However, *Acidobacteria* was less abundant at the site with the lowest pH (i.e., the black spruce site as well). Instead, the relative abundance of *Acidobacteria* appeared to share a negative relationship with the soil C content, which is consistent with previous findings [10]. This suggests that, in this case, other parameters such as soil nutrients could play a more important role than pH in shaping bacterial communities, but it is also possible that the soil pH range (~2.9–4) was too narrow to observe a significant impact on *Acidobacteria* abundance.

In addition to its influence on soil chemistry, some tree species establish host-specific associations with soil microbes. Fungi—more particularly, saprotrophs and ECM—preferentially colonize the bulk or rhizosphere soil of particular tree stands [66], and are often endemic to specific bioregions [67]. On the other hand, bacteria are primarily influenced by soil pH and appear to be less tree-species-specific than fungi [63]. These differences in environmental drivers, colonization strategies, and dispersion abilities between the two micro-organism kingdoms could explain why we observed a more pronounced site effect on fungi than on bacteria (Table 1). Information regarding the preference of fungal taxa for specific trees is relatively scarce, but our results suggest that Ascomycota may favour deciduous over coniferous trees and Basidiomycota may favour coniferous over deciduous trees. This is supported by Urbanová et al. (2015) [63], who found that soil in coniferous tree stands (i.e., *Picea* spp. and *Pinus* spp.) displayed a higher abundance of Basidiomycota. In addition, reduced precipitation can lead to decreased Ascomycota abundance while increasing Basidiomycota's abundance in forest soils [68], which was observed in our study, as the ASV abundance of both phyla followed a similar pattern of variation according to mean annual precipitation.

The microbial communities also differed between the L, F, and H soil horizons, with more pronounced contrasts between the litter and deeper horizons (Figures 1 and 2), which has also been reported in previous works [69]. Litter displayed higher relative

abundances of *Proteobacteria*, *Bacteroidetes*, Ascomycota, and Basidiomycota; lower relative abundance of *Acidobacteria*; and higher microbial diversity than the F and H horizons. Similar results regarding both soil bacteria and fungi have been reported in Central European forests [63,70,71]. However, we found a contradictory distribution pattern of *Actinobacteria*, the abundance of which decreased between the litter and the F and H horizons. This was most likely due to the dominant genus, *Acidothermus*, accounting for ~17% of the overall bacterial community, which became significantly more abundant in deeper horizons. These shifts in microbial communities within the soil may be related to their preferential substrates, as soil horizons have contrasting chemical composition (e.g., organic C and lignin concentration [64]) due to the vertical stratification of decomposition processes. For example, copiotrophic bacteria and saprotrophic fungi are more abundant in litter, while oligotrophic bacteria and ECM are more abundant in deeper horizons [70]. This was confirmed by our results, which showed a higher abundance of the saprotrophic genera *Cladophialophora*, *Luellia*, *Pezoloma*, and *Venturia* in litter and a tendency of higher abundance of the ECM genera *Cortinarius*, *Piloderma*, and *Russula* in F-H horizons, closer to tree roots. In addition, *Leptothrix*, which is affiliated with the copiotrophic phylum *Proteobacteria*, was more abundant in litter, and *Acidibacter*, which is affiliated with the oligotrophic phylum *Acidobacteria*, were more abundant in the F-H horizons. Because members of *Actinobacteria* were more abundant in the litter while other preferred deeper horizons, we suggest that nutritional and growth strategies are also genus-dependent, with genera such as *Galbitalea* and *Jatrophihabitans* being copiotrophic, but *Acidothermus* being oligotrophic. However, this bacterial functional classification remains highly approximative, and these assumptions need to be confirmed by further work.

### 4.2. Moderate Effects of Long-Term Increased N Deposition on Forest Soil Microbial Communities

In our study, simulated N deposition had significant but small effects on soil microbes across the three forest sites (Table 1). Contrary to our prediction, we found that N addition did not significantly change the soil microbial diversity or composition (Figures 5 and 6). This also strongly contrasts with recent N fertilization experiments, which have reported decreasing bacterial diversity in amended forest soils [15,72], and a meta-analysis which revealed the global deleterious impacts of N addition on soil fungal diversity [30]. Moreover, we expected that, following N addition, copiotrophic bacteria would become more abundant than oligotrophic bacteria. We observed this shift at the ASV level, but to an extremely small extent, as only eight ASVs and three assigned genera were affected by N addition (Figure 4a). We also predicted that, under higher N conditions, fungal communities would switch from white-rot-dominated to brown-rot-dominated. We did not observe such a pattern, and instead found that the abundance of a few white-rot fungi genera (*Luellia* and *Sistotremastrum*) increased.

The often-reported deleterious impacts of simulated higher N deposition on soil microbes are most likely due to the well-known acidification of soil and subsequent changes in soil chemistry caused by ammonium nitrate application [12]. However, in previous studies, soil analyses performed on the same experimental plots at the balsam fir and the black spruce sites [73], as well as at the sugar maple site [43], revealed that soil pH and N concentration showed either small or no changes, suggesting that the sites did not become N-saturated, which is in agreement with our pH and N analysis (Table S2). Conversely, moss and foliar N concentrations increased in coniferous N-amended plots, suggesting that N sequestration in vegetation prevents N accumulation in soils and has consequent drastic impacts on below-ground microbial communities. Other processes leading to the mitigation of N accumulation in soils are leaching, denitrification, and volatilization, three well-known pathways of N loss in forest ecosystems [74]. High precipitation volume during the growth season in the eastern Canadian forest (i.e., period of N treatment application as well) may have minimized the effects of N addition by increasing organic and inorganic N (especially $NO_3^-$) leaching in soils. This was observed by N fertilization experiments in Asia, where heavy precipitation may have alleviated the effects of N on soil bacteria [72,75]. However,

recent results showed that the leaching of amended N is very low at the balsam fir and black spruce sites [73], suggesting that leaching did not significantly affect the efficiency of N treatments. It is also unlikely that denitrification had a significant effect on the N fertilization results, as it is generally considered to be low in boreal forests [76]. Ammonia volatilization at the soil surface following N treatment certainly occurred [77], but given the duration and the application rate of the treatments, it probably had an overall minimal contribution to N loss.

It is also possible that during the 16 years of the experiment, some micro-organisms had the time to adapt to the new environmental conditions caused by higher N deposition, and that the soil microbiome became partially resilient [78]. However, complementary work is necessary to verify whether the forest soil microbiome functions remained unchanged (i.e., functional redundancy). Overall, the relatively low N application rate range used in our study (<90 kg ha$^{-1}$ yr$^{-1}$), combined with rapid N uptake by plants, most likely explain the moderate effects of N treatment on the soil microbiome which were observed.

*4.3. Potential Consequences for Northern Temperate and Boreal Forest Ecosystems*

We report that N addition had a very minimal impact on bacteria, but a more pronounced impact on fungi, especially on the Basidiomycota and Ascomycota phyla. The ASV abundance of saprotrophic Agaricomycetes *Membranomyces*, *Mycena*, *Serendipita*, and, more specifically, the white-rot genera *Luellia* and *Sistotremastrum*, significantly increased under higher N conditions (Figure 4b). As fungal biomass was shown to increase after N fertilization at sites with low N deposition [79], it is possible that, in our experiment, N similarly alleviated fungal N limitation and subsequently promoted the abundance of these genera. Basidiomycota and Ascomycota have crucial roles in organic matter decomposition, and the known members of Agaricomycetes are mostly wood decomposer fungi. Therefore, our results suggest that higher N deposition could, to a small extent, increase the abundance of decomposer taxa, leading to an acceleration of organic matter decomposition and ultimately reducing C accumulation in northern temperate and boreal forest soils. However, this contradicts previous work performed in temperate and boreal forests [80–82]. This could be explained by the study sites' N limitations not being overcome, potentially due to rapid N mobilization by the vegetation (mentioned previously) increasing the oxidative enzymatic activity and growth of white-rot fungi [25]. In this case, overall decomposition would be promoted by N fertilization in N-limited forests.

Along with saprotrophs, N treatments also impacted ECM fungi. While the abundances of a few genera were negatively affected by N addition (i.e., *Russula* and *Clavulina*), overall ECM abundance appeared to increase under high N treatment. This shift was also observed at the black spruce site after 8 years of N addition [17]. Fertilization could have stimulated tree growth (already visible at the black spruce site after 8 years of treatment [73]) and the expansion of the root system [83], and, consequently, promoted tree associations with ECM. This implies that long-term increased N deposition could have an impact not only on the decomposition process, but also on nutrient transfer from the soil and litter to plants by ECM.

Increased N deposition acts on soil ecosystems by decreasing pH and modifying C and N concentrations, which can induce shifts in understorey plant communities and further explain the changes in microbial communities following N addition. For example, we noticed a substantial reduction in the moss cover of the black spruce units receiving high N treatment. This has been observed in other N fertilization experiments as well [84,85], and may be the result of NH$_4^+$ toxicity or increased competition between moss and vascular plants drastically impairing moss growth in northern forest ecosystems [84,86]. This modification of moss cover density could indirectly impact soil microbes by reducing temperature and moisture regulation and changing nutrient flux towards the soil [87,88]. In our study, the reduction in moss cover and global vegetation modification induced by N addition may explain the noticeable changes in fungal communities compared to bacteria, especially for ECM, which share close relationships with plants. However, further work

is needed to fully elucidate the influence of trees and understorey vegetation on the soil microbiome following increased N deposition.

## 5. Conclusions

In this study, we report that 16 years of periodic N addition had minimal impacts on soil bacteria and fungi in the northern temperate and boreal forests of eastern Canada. Treatments corresponding to 48 and 160 years of accelerated N deposition had no significant effects on microbial relative abundance or diversity. Several meta-analysis studies have indicated that increased N deposition has global detrimental impacts on the soil microbiome [16,89]. Instead, our work suggests that, in N-limited high-latitude forests, supplementary N is rapidly taken up by vegetation, therefore preventing N accumulation in soils and microbiome perturbations. Moreover, additional N could promote vegetation and root growth and ultimately favor the establishment of associations with ECM, explaining their overall higher ASV abundances in our experimental sites. Nonetheless, our work provides evidence that the eastern Canadian forest soil microbiome might be more resilient to the predicted increase in anthropogenic N deposition than previously thought. On the other hand, we demonstrated that the site and soil horizons, characterized by specific environmental conditions, strongly influenced the soil microbial communities. This indicates that, even if N deposition might have a moderate impact on soil bacteria and fungi, climate change could have a much more significant effect on high-latitude forest soil microbiomes. In this context, beyond investigating the microbial community's taxonomic identity, the impacts of increased N deposition and global change on soil microbiome functions should be further explored in order to better predict the future of northern forests.

**Supplementary Materials:** The following supporting information can be downloaded at: https://www.mdpi.com/article/10.3390/f14061124/s1. Supplementary File S1: R script for 16S rRNA and ITS amplicon sequencing bioinformatic and statistical analyses. Supplementary File S2: Dataset used for 16S rRNA and ITS amplicon sequencing statistical analyses in R. Supplementary File S3: Dataset comprising soil pH, N concentration, and C:N, measured in 2013 and 2021, at the three experimental sites and used for statistical analyses in R. Table S1: Environmental characteristics of the experimental sites selected for this study (from Ste-Marie and Houle, 2006). Table S2: Results of one-way ANOVA and Kruskal-Wallis tests showing soil N concentration, C:N, and pH statistical differences between N addition treatments in F and H horizons at the sugar maple, balsam fir, and black spruce experimental sites. For all statistical tests, degrees of freedom are equal to two. Stars indicate statistically significant $p$ values. Figure S1: Soil bacterial (a) and fungal (b) ASVs differential abundance analysis between study sites in control plots. Points represent ASVs with abundance differing significantly ($p < 0.05$) between treatments. Colors were assigned to phyla. Figure S2: Soil bacterial (a) and fungal (b) ASVs differential abundance analysis between L, F, and H horizons in control plots at the sugar maple, the balsam fir, and the black spruce sites. Points represent ASVs with abundance differing significantly ($p < 0.05$) between treatments. Colors were assigned to phyla. Figure S3: Soil pH (at the sugar maple site), N concentration, and C:N (at the balsam fir site) under control, low N, and high N treatments measured in the F horizon in 2013. Letters indicate statistical differences between treatments.

**Author Contributions:** Conceptualization, D.H. and J.-D.M.; methodology, D.H., S.W.K. and J.-D.M.; software, M.R.; validation, D.H. and S.W.K.; formal analysis, M.R.; investigation, M.R., R.K. and S.L.; resources, D.H., S.W.K. and D.K.; data curation, M.R., R.K. and S.L.; writing—original draft preparation, M.R. and R.K.; writing—review and editing, M.R. and D.H.; visualization, M.R.; supervision, D.H.; project administration, D.H.; funding acquisition, D.H. All authors have read and agreed to the published version of the manuscript.

**Funding:** This research was supported by the Science and Technology Branch of Environment and Climate Change Canada, Ouranos, and the Mitacs Accelerate program.

**Data Availability Statement:** The R script and database used in this study are publicly available on Figshare (https://figshare.com/authors/Marie_Renaudin/11938571, accessed on 27 April 2023). All raw sequence files were deposited into the NCBI BioProject database under accession number PRJNA880910 (https://www.ncbi.nlm.nih.gov/bioproject/PRJNA880910, accessed on 15 September 2022).

**Acknowledgments:** We would like to thank B. Toussaint, J. Martineau, J. Gagné, and M. St-Germain for field assistance.

**Conflicts of Interest:** The authors declare that they have no conflicts of interest.

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
