# Peer review of "Long-Term Simulated Nitrogen Deposition Has Moderate Impacts on Soil Microbial Communities across Three Bioclimatic Domains of the Eastern Canadian Forest"

_forests, doi:10.3390/f14061124_

Round 1

Reviewer 1 Report

The manuscript "Long-term simulated nitrogen deposition has moderate impacts on soil microbial communities across three bioclimatic domains of the eastern Canadian forest" by Khlifa et al., aims to investigate the effects of increased N deposition on soil bacterial and fungal communities across three bioclimatic domains receiving natural N deposition. 

This is a well-written, well-organized manuscript. The methodology is sound, and the statistical analyses are robust. The research on multiple soil horizons is very relevant and interesting to the readers. 

I have only two suggestions; the first one, the authors need to tune down the affirmation of "contrasting habitat conditions" (L430) since even if they are studying different bioclimatic domains, the contrast is not high, such as the expected in larger scales. Changing the wording of "contrasting" for "different" should be enough. They recognize the narrow range of some variables, e.g., pH. 

The second suggestion is to incorporate other vias of N loss in the discussion, e.g., lixiviation and volatilization.

Author Response

We thank you for your review. Please see the attachment for the response to your comments.

Reviewer 2 Report

General evaluation of the manuscript

The present study assessed the effects of excessive nitrogen deposition on the microbiome of forest soils in eastern Canada. The experiment was conducted on three sites that were fertilised with nitrogen at rates equivalent to 3 times and 10 times the natural atmospheric deposition of this element. Changes in microbial community (bacteria and fungi) were assessed after 16 years of the simulation experiment using the 16srRNA sequencing method. A total of 78 samples were sequenced, resulting in a large sequence database, which was subjected to bioinformatics analysis. It was shown that long-term increases in N deposition had no effect on microbial diversity, with only minor changes in the composition of the bacterial and fungal communities observed.  The research was well planned, the methods are accurately described, the results correctly analysed and well discussed. The work provides valuable and comprehensive information on the diversity and structure of the forest soil microbiome.          

Author Response

We thank you for your review. Please see the attachment for the response to reviewers' comments.

Reviewer 3 Report

In this paper, the effects of N deposition on forest ecosystems were analyzed from the perspective of microorganism, and some interesting results were obtained. The conclusion of the article is very good, which will promote the development of the subject. I think this is very good for publication in Forests. I have a few suggestions for your reference.

1. Please provide appropriate information on global or regional N deposition to facilitate understanding of the importance of the research topic.

2. It is suggested to appropriately supplement the basic characteristics of the three study areas.

3. It is suggested to further clarify the key scientific questions in this paper.

Author Response

(The authors gave the same response as above.)
